# Coloration and Multi-Functionalization of Polypropylene Fabrics with Selenium Nanoparticles

**DOI:** 10.3390/polym13152483

**Published:** 2021-07-28

**Authors:** Tarek AbouElmaaty, Shereen A. Abdeldayem, Shaimaa M. Ramadan, Khaled Sayed-Ahmed, Maria Rosaria Plutino

**Affiliations:** 1Department of Material Art, Galala University, Galala 43713, Egypt; 2Department of Textile Printing, Dyeing and Finishing, Faculty of Applied Arts, Damietta University, Damietta 34512, Egypt; shereen.abdeldayem.82@gmail.com (S.A.A.); designer_shemo@yahoo.com (S.M.R.); 3Department of Agricultural Chemistry, Faculty of Agriculture, Damietta University, Damietta 34512, Egypt; dr_khaled@yahoo.com; 4Stituto per lo Studio dei Materiali Nanostrutturati, Consiglio Nazionale delle Ricerche, Vill. S. Agata, 98166 Messina, Italy; plutino@pa.ismn.cnr.it

**Keywords:** selenium nanoparticles, polypropylene, coloration, antibacterial, conductivity, UV protection

## Abstract

In this study, we developed a new approach for depositing selenium nanoparticles (SeNPs) into polypropylene (PP) fabrics via a one-step process under hydrothermal conditions by using an IR-dyeing machine to incorporate several functionalities, mainly coloration, antibacterial activity and ultraviolet (UV) protection. The formation, size distribution, and dispersion of the SeNPs were determined using X-ray diffraction (XRD), ultraviolet-visible (UV/Vis), transmission electron microscopy (TEM) and the color strength, fastness, antibacterial properties, and UV protection of the treated fabrics were also explored. The UV-Vis spectra and TEM analysis confirmed the synthesis of spherical well-dispersed SeNPs and the XRD analysis showed the successful deposition of SeNPs into PP fabrics. The obtained results demonstrate that the SeNPs-PP fabrics is accompanied by a noticeable enhancement in measurements of color strength, fastness, and UV-protection factor (UPF), as well as excellent antibacterial activity. Viability studies showed that SeNPs-PP fabrics are non-toxic against wi-38cell line. In addition, the treated SeNPs-PP fabrics showed an increase in conductivity. The obtained multifunctional fabrics are promising for many industrial applications such as the new generation of curtains, medical fabrics, and even automotive interior parts.

## 1. Introduction

Numerous washable or disposable healthcare and hygiene textile products are used in hospitals either for the protection of staff and patients (drapes, beddings, masks, uniforms, wound dressings, bandaging materials, etc.) [1]. Due to the large surface areas, textiles have superior abilities to retain warmth, moisture, and nutrients from spillages and exudates, making them ideal substrates for microorganisms to grow on [2]. Some studies have suggested that healthcare textiles can act as reservoirs and vehicles for the spread of microorganisms in hospitals [3]. With the development of nanotechnology, some inorganic nanoparticles (NPs), such as silver, copper, and zinc, have been identified as promising candidates in combating pathogenic microorganisms [4,5,6,7,8,9]. Nanotechnology-driven therapies with metal and metal oxide nanoparticles (NPs) are emerging as a promising alternative to antibiotics. The high reactivity of these NPs, due to large surface-to-volume ratio, results in intrinsic targeted antimicrobial efficiency even when they are applied in small amounts [10]. Proven activity of metal and metal oxide NPs against wide range of micro-organisms including bacteria, fungi, viruses, and other eukaryotic microorganisms [11] was inspiring for researchers to immobilize them onto textiles [12,13,14,15]. Selenium nanoparticles (SeNPs) has become the new research target because they are found to possess excellent bioavailability, low toxicity, and contribute to a wide spectrum of health promotion, as well as disease prevention and treatment activities [16]. Biswas et al. [17] prepared silver or selenium nanoparticles on polymeric scaffolds and compared the cytotoxicity of the scaffolds towards mouse fibroblasts using an indirect contact method; the results indicated that the Ag-loaded scaffolds showed high cytotoxicity, while the Se-loaded scaffolds were not toxic to the cells. The low cytotoxicity of SeNPs indicates the great potential for use in biomedical applications [3]. The research on SeNPs as antimicrobial agents is still limited. Several studies have pointed out the ability of SeNPs to exhibit anticancer [18], antioxidant [19], antibacterial and anti-biofilm [20] properties. So far, remarkable antimicrobial activity of these nanoparticles has been evidenced against pathogenic bacteria, fungi, and yeasts [21,22,23], which inspired researchers to immobilize them onto textiles.

Polypropylene (PP) fabric has excellent physical and mechanical properties [24]. It is a hydrophobic fabric, and several surface modification techniques are adopted to improve wetting, adhesion to polymer surfaces by introduction of a variety of polar groups [25,26,27]. PP advantages include a great supply, good process, low energy demand, low cost and high chemical stability [28]. There is a great demand for antibacterial PP fabric to be used in different medical applications [29,30]. To our knowledge, there are no reports dealing with coloring PP by using SeNPs without any additives and studying the antibacterial and UV-protection properties, cytotoxicity, and electrical conductivity of the resulted SeNPs-PP fabrics. In this study, SeNPs are prepared in one step process under a simple redox system based on the method mentioned by Abou Elmaaty et al. [31] followed by application of the SeNPs to PP fabric in one-step process under hydrothermal conditions. The characteristics of SeNPs in solution phase were studied by UV-visible spectrophotometer (UV/Vis), transmission electron microscopy (TEM) and X-ray diffraction (XRD). Moreover, scanning electron microscopy (SEM), colour characteristics and antibacterial and UV-protection properties, cytotoxicity and electrical conductivity of the SeNPs-PP fabrics were also evaluated. In summary, we developed a simple, green, and feasible route to produce green coloration of PP-based fabric for multifunctional applications. The low cytotoxicity of SeNPs and antibacterial properties indicates their great potential for use in biomedical applications.

## 2. Materials and Methods

### 2.1. Fabric and Chemicals

Polypropylene fabric (100%) was supplied by Shikisen-sha Company (Osaka, Japan) with crystallinity (50.6%), melting enthalpy (105.8 J/g) [32], density (0.91 g/c.c), moisture regain (0%) and tenacity (3.5–8.0 g/den). Sodium hydrogen selenite, ascorbic acid and polyvinylpyrrolidone (PVP) were purchased from *LobaChemie, India*. Other chemicals were commercial grade.

### 2.2. Green Synthesis of Selenium Nanoparticles (SeNPs)

SeNPs were synthesized via a redox reaction based on the method reported by AbouElmaatyet al. with an improved modification [31]. PVP (6g) was dissolved in 100 mL of sodium hydrogen selenite solution at a concentration of 100 mmol/L. Then, ascorbic acid was added to the mixture at the same concentration and volume ratio of 1:1 under magnetic stirring. The solution changed from colorless to orange to dark orange, indicating the formation of SeNPs [33]. Moreover, the prepared SeNPs colloidal solution at the concentration of 50 mmol/L was used in this treatment process.

### 2.3. Treatment Method

The PP fabrics were treated using an infrared dyeing machine. The machine includes 12 beakers fixed in a rotating carrying wheel. Heating was obtained by IR, cooling by air, and automation by the microprocessor programmer DC4 F/R. The highest temperature used in this device was120 °C, the highest rate of heating was 2 °C per minute, and the highest rate of cooling was 6 °C per minute. First, a solution of SeNPs (50 mmol/L) was prepared. The PP fabrics were then immersed in the solution with liquor ratio (LR) of 1:50. Next, the device was set at the following treatment temperatures periods. The treatment was performed at different temperatures (70 °C, 100 °C and 120 °C) and different periods (1, 2 and 3 h). The treated PP fabrics were rinsed with distilled water and allowed to dry at room temperature after the treatment. The obtained fabrics were coded with SeNPs-PP fabrics.

### 2.4. Characterization

#### 2.4.1. Characterization of Selenium Nanoparticles (SeNPs)

##### Transmission Electron Microscopy (TEM) Analysis

The size and morphology of SeNPs were characterized using JEM-2100 Transmission Electron Microscope with an acceleration voltage of 200 kV. A drop of colloidal solution containing SeNPs was dripped onto a carbon coated copper grid and dried at room temperature for TEM analysis.

##### X-ray Diffraction (XRD) Analysis

X-ray diffraction (XRD) analysis was conducted for synthesized SeNPs and SeNPs-PP fabrics using an X-ray diffractometer system (Bruker D8 ADVANCE, Karlsruhe, Germany). While SeNPs solution was dried at 130 °C until completely dryness before the XRD analysis.

##### UV/Vis Spectroscopy Analysis

SeNPs were further characterized via ultraviolet-visible spectrophotometer (Alpha-1860, Indianapolis, IN, USA), and their formation was confirmed by the maximum absorption peak which attributed to their surface plasmon resonance.

#### 2.4.2. Characterization of Poly Propylene (PP) Fabrics

##### Scanning Electron Microscopy (SEM) Analysis

The surface morphology of blank PP and SeNPs-PP fabric was characterized via scanning electron microscope (JEOL JSM-6510LB with field emission gun, Tokyo, Japan). The deposition of SeNPs into PP fabrics was confirmed using surface energy dispersive x-ray (EDX) analysis unit (EDAX AMETEK analyzer) attached to SEM device.

##### Raman Spectroscopy Analysis

The types of bonds present in the blank PP and SeNPs-PP fabrics were determined using confocal Raman microscope (Jasco NRS-4500, Tokyo, Japan) which covered the range from 200 to 4000 cm^−1^. Raman data acquisition and processing were performed using Jasco spectroscopy suite software.

##### Colorimetric Study

The colorimetric parameters such as lightness (L*), redness-greenness (a*), yellowness-blueness (b*) and the color uptake which is expressed as the color strength (K/S) of the obtained SeNPs-PP fabrics were determined using a spectrophotometer (CM3600A; Konica Minolta, Tokyo, Japan). K/S values were evaluated at the wavelength of maximum absorption (λ_max_) of the color’s reflectance curve at390 nm.

##### Exhaustion of SeNPs into PP Fabric

The treatment colloidal solution was sampled before and after treatment to measure the exhaustion of SeNPs. The absorbance of the SeNPs colloidal solution was measured using an UV/Vis spectrophotometer (Alpha-1860, Indianapolis, IN, USA).

##### Physical Properties of SeNPs-PP Fabric

The fastness of the SeNPs-PP fabrics determines the fixation of SeNPs into the fabric. They were determined using AATCC (61-1972), (8-1972), and (16A-1972) [34] tests for washing, rubbing and lightfastness, respectively. The tensile strength tests of the PP and SeNPs-PP fabrics were performed using a universal testing machine (Tinius Olsen EN ISO 13934-1;1999-model H25KT) [35]. Additionally, the durability to washing was evaluated according to AATCC61(2A)-1996 test [36] after five washing cycles.

##### Cytotoxicity Test of SeNPs-PP Fabric

The cytotoxicity of SeNPs-PP fabric was tested against wi-38 cell line. This type of cells is diploid human cell line, including fibroblasts from lung tissue of a 3-month-gestation female fetus. The SeNPs-PP fabric treated at the optimum conditions was sterilized, cut, and plated on the bottom surface of a six-well tissue culture plate. The plate was inoculated with 1 × 105 cells/mL (100 µL/well) and incubated at 37 °C for 24 h. Additionally, the growth medium was decanted, and the cell monolayer was washed twice with washing media. Cells were checked for any physical signs of toxicity. Moreover, the tissue was picked up and 20 µL 3-(4,5-dimethylthiazol-2-yl)-2,5-diphenyltetrazolium bromide dye (MTT) prepared in phosphate buffer saline (BIO BASIC CANADA INC, Markham, Ontario, Canada) was added to each well at a concentration of 5mg/mL and shaken for 5 min. The wells were incubated at 37 °C and 5% CO2 for 1–5 h. After dumping the media, the formazan as MTT metabolic product in the dry plate was resuspended in 200 µL dimethyl sulfoxide and shaken for 5 min. Then, the optical density was read at 560 nm, while the background was measured at 620 nm and subtracted.

##### Antibacterial Activity

The antibacterial activity of the SeNPs-PP fabric was evaluated using AATCC (147-2004) test [37]. Antibacterial tests were carried out against G+ve bacteria (*Staphylococcus aureus* and *Bacillus cereus*) as well as G-ve bacteria (*Escherichia coli* and *Pseudomonas aeruginosa*) and the growth inhibition zone (mm) was determined.

##### UV-Protection Properties

The UV protection factor (UPF) and UV-blocking activities of the SeNPs-PP fabric were determined using Standards Australia and Standards New Zealand (AS/NZS) 4399:1996 tests and UV protection properties were expressed as good, very good, or excellent for UPF values of 15–24, 25–39, and >40, respectively.

##### Electrical Conductivity Measurement

The electrical conductivity of both the PP and SeNPs-PP fabrics were measured using LRC-bridge (Hioki model 3531zHi Tester, Nagano, Japan).

## 3. Results and Discussion

### 3.1. Characterization of SeNPs

#### 3.1.1. Transmission Electron Microscopy (TEM) Analysis

TEM micrographs confirmed the formation of spherical Se-NPs in the range of 31–79 nm. Furthermore, the synthesized SeNPs were well-dispersed with no aggregation and deformation as shown in Figure 1.

Histogram bins are10-nm wide and centered at 35, 45, 55, 65 and 75 nm. All SeNPs with diameters from 40 nm to 50 nm were considered to have a size of 45 nm.

Furthermore, TEM was used to examine the adsorption of SeNPs as shown in Figure 1. The TEM images illustrated that SeNPs were sufficiently monodispersed and adsorbed on the SeNPs-PP fabric surface.

#### 3.1.2. X-ray Diffraction (XRD) Analysis

XRD confirmed the formation of SeNPs and their deposition into the treated PP fabric based on the crystallinity of SeNPs. As illustrated in Figure 2, SeNPs in a colloidal solution or into the PP fabric surface were highly crystalline. Additionally, the diffraction peaks at 24.28°, 29.24°, 43.64° and 64.28° were corresponding to 100, 101, 102 and 210 crystal planes, respectively, based on the JCPDS 86-2246 international database [38]. Moreover, the peaks at 13,66, 16.56, 18.18, and 25.38 can be observed for PP fabric, corresponding to the planes of (110), (040), (130), and (060), respectively [39].

#### 3.1.3. UV/Vis Spectroscopy Analysis

The formation of SeNPs was confirmed from the UV/Vis spectra based on their SPRs. The solution changed from colorless to orange to dark orange, indicating the complete reduction of sodium hydrogen selenite to SeNPs [40].

As shown in Figure 3, the SeNPs colloidal solution showed an absorption peak at 263 nm, confirming the formation of the spherical SeNPs [41].

### 3.2. Characterization of Poly Propylene (PP) Fabrics

#### 3.2.1. Scanning Electron Microscopy (SEM) Analysis

The SEM micrographs of the PP fabric revealed that the surface was clear with clean scales and typical fibrous structure as displayed in Figure 4. On the other hand, the SEM micrographs of the SeNPs-PP fabric show a coated layer of SeNPs into the PP fabric. Additionally, SeNPs were well distributed into the fabric.

The chemical elements found on the surface of the treated PP fabric were analyzed using EDX. The peaks around 1 and 11 Kev are attributed to SeNPs. The carbon and oxygen peaks were belonged to the native PP fabric. However, other elements were monitored at low concentration, such as Si, Ca, and Fe. Those traces of elements can be attributed to using IR-dyeing technique [42]. In addition, the Sulphur element was monitored at low concentration because EDX is an elemental detection technique with a certain small error. Both SEM micrographs and EDX analysis confirmed the deposition of SeNPs into the PP fabric surface as displayed in Figure 4.

#### 3.2.2. Raman Spectroscopy

Raman analysis revealed the chemical bonds inside the PP fabric and between the SeNPs and PP surface. This analysis is important to compare the chemical structure of Se-NPs-PP fabric and PP fabric as shown in Figure 5. The peaks at 2960 as well as 2888 cm^−1^ were corresponding to the asymmetric stretch of methyl group [43]. While the peak at 984 cm^−1^ was associated with asymmetrical stretching of C-C bond [44]. Furthermore, Raman analysis revealed the presence of SeNPs into the SeNPs-PP fabric. The treated fabric showed an obvious peak at 236 cm^−1^, corresponding to the symmetric stretching of SeNPs [45]. On the other hand, no peak was observed in this region in the case of PP fabric without SeNPs.

#### 3.2.3. Colorimetric Study

The color parameters of the SeNPs-PP fabric were analyzed using a Konica Minolta spectrophotometer (CM-3600A). Figure 6 and Table 1 show the L*a*b* values of the fabric, where (L*) values represent color lightness, (a*) is the red/green coordinate, and (b*) is the yellow/blue coordinate [46]. These values indicated that SeNPs-PP fabric is darker according to the color lightness values L*, less red and less yellow according to a*, b* values, respectively.

##### Effect of Treatment Time on Color Strength (K/S)

The relationship between the K/S value of the SeNPs-PP fabric and treatment time (1, 2, and 3 h) is shown in Figure 7. Notably, the *K*/*S* value of PP fabric treated with SeNPs (concentration of 50 mmole/L, L.R 1:50 and a temperature of 100 °C) increased with an increase in the treatment time. The increase in K/S value reflected the positive effect that increasing the treatment time had on the uniformity of PP adsorption of the SeNPs, and on the uniformity of the penetration and diffusion of the SeNPs into the fabric; these effects in turn contributed to an increase in the SeNPs uptake by the fabric [47,48]; which is indicated by the highest K/S value observed at treatment time of 3 h.

##### Effect of Treatment Temperature on Color Strength (K/S)

Figure 8 shows the relationship between treatment temperature and the color uptake (K/S) of the SeNPs-PP fabric with a (concentration of 50 mmol/L, L.R 1:50 and a treatment time of 3h). K/S increased linearly with an increase in temperature from 70°C to 120°C and increased considerably at low temperatures until the color approached to an equilibrium point above 100°C. The molecular structure opens, which facilitates the uptake of NPs as the temperature increases. Hence, a high K/S value is obtained. This can be attributed to an increase in temperature, which improves the macromolecular chains of PP. Moreover, large pores and/or channels suitable for NPs penetration and diffusion are formed. Hence, the optimum temperature was set at 100 °C [49,50,51,52].

#### 3.2.4. Exhaustion of SeNPs into PP Fabric

The treatment solution was sampled before and after treatment to measure the SeNPs exhaustion. Moreover, the absorbance of SeNPs solution was measured by using UV/VIS spectrophotometer-model: Alpha-1860. Figure 9 showed the absorbance of SeNPs concentration before and after exhaustion by PP fabric in the wavelength ranges from 200 to 700 nm. The absorption spectrum of SeNPs-PP fabric before exhaustion shows a sharp absorption band at 263 nm, indicating the presence of SeNPs. After exhaustion, remarkable decrease in the absorbance of the treatment solution can be attributed to the low ratio of SeNPs as it was absorbed by PP fabric.

#### 3.2.5. Physical Properties of SeNPs-PP Fabric

The fastness of the SeNPs-PP fabric treated under optimum conditions was evaluated mainly in washing, rubbing, and lightfastness. In addition, the tensile strength (elongation and maximum force) was also evaluated. According to the results listed in Table 2, it can be concluded that there is a decrease in elongation and a little increase in maximum force of SeNPs-PP fabric without causing a significant damage to the structure of the yarn indicating no significant change between PP and SeNPs-PP fabrics. The washing and rubbing fastness were excellent, even after five washing cycles. Additionally, the light fastness of SeNPs-PP fabric was also found to be in the field of good to very good, this indicates that the fixation of SeNPs onto PP fabric may be attributed to the generation of metal chelates [53].

#### 3.2.6. Cytotoxicity of the SeNPs-PP Fabric

The cytotoxicity of the SeNPs-PP fabric was evaluated against healthy human cells (wi-38) by the MTT assay. The viability of cells and the cytotoxicity of SeNPs-PP fabric were evaluated against wi-38 cell lines. The viability of cells of SeNPs-PP fabric was 99.46% of that of negative control. Whereas over 70% is the mean relative cell viability [54]

#### 3.2.7. Antibacterial Activity of the SeNPs-PP Fabric

Table 3 lists the antibacterial activity of the SeNPs-PP fabric using four bacterial strains, (*Staphylococcus aureus* and *Bacillus cereus*) as Gram-positive bacteria and (*Escherichia coli* and *Pseudomonas aeruginosa*) as Gram-negative bacteria. Tetracycline and ciprofloxacin were used as standard drugs. The results reveal that the SeNPs-PP fabric exhibits excellent antibacterial activity against *Escherichia coli*, *Bacillus cereus*, as well as *Staphylococcus aureus* and very good antibacterial activity against *Pseudomonas aeruginosa*, which is indicated by a clear zone diameter of bacterial colonies. The obtained results indicate the presence of broad-spectrum antibacterial activity. The mechanism of action of SeNPs on bacteria is still unclear. In this study, we suggest the following phenomena: (a) the release of ions along with physical interaction with the bacterial cell wall peptidoglycan layer damages the double-stranded structure of DNA. (b) the formation of reactive oxygen species and inhibition of DNA replication. (c)nanoparticles can be in better contact with bacterial or fungal cells than colloidal form [53].

#### 3.2.8. UV-Protection Properties of the SeNPs-PP Fabric

Table 4 lists the results of the UV light protection characterization. The SeNPs-PP fabric effectively blocked UV radiation; based on the AATCC test criteria: if UPF of any fabric is more than 40as the fabric is a UV-defensive material [55].

#### 3.2.9. Electrical Conductivity Measurement

The treatment of PP fabrics with SeNPs led to a slight increase in the electrical conductivity of PP fabrics. Samples treated with the optimum concentration of SeNPs showed EC value of 5.84 × 10^−11^Ω^−1^ Cm^−1^ compared to untreated fabric which had EC value of 1.06 × 10^−11^ Ω^−1^ cm^−1^.

## 4. Conclusions

In this paper, we propose a novel approach for coloring and incorporating new functionalities to PP fabrics via one-step process using SeNPs. PP fabrics were colored from light to dark orange depending on the treatment time and concentration of sodium hydrogen selenite. The obtained results show that the deposition of SeNPs into PP fabric is accompanied by a considerable improvement in UV-protection. The obtained colored fabrics effectively blocked UV radiations, providing excellent UV-protection. Additionally, the treated fabrics exhibited outstanding washing, rubbing and light fastness. Moreover, the colored PP fabric showed excellent antibacterial activity against *Staphylococcus aureus*, *Bacillus cereus*, and *Escherichia coli* and very good antibacterial activity against *Pseudomonas aeruginosa* compared with the standard drugs such as tetracycline and ciprofloxacin. The tensile strength of the colored fabrics increased slightly accompanied by a slight decrease in elongation. This novel, and economical approach can be employed in the industry for coloration and multifunctionalization of PP fabrics instead of traditional dyeing and finishing processes.

## Figures and Tables

**Figure 1 polymers-13-02483-f001:**
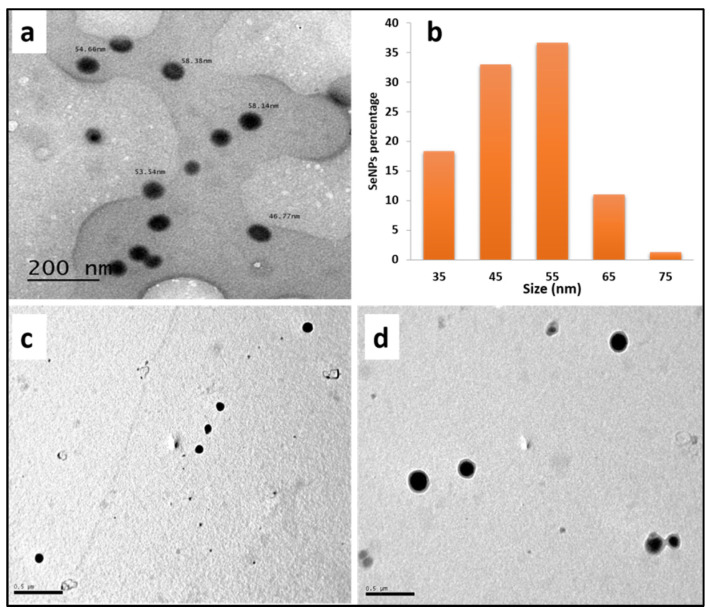
(**a**) TEM image (**b**) size distributions histogram of prepared SeNPs, (**c**,**d**) SeNPs-PP fabric surface.

**Figure 2 polymers-13-02483-f002:**
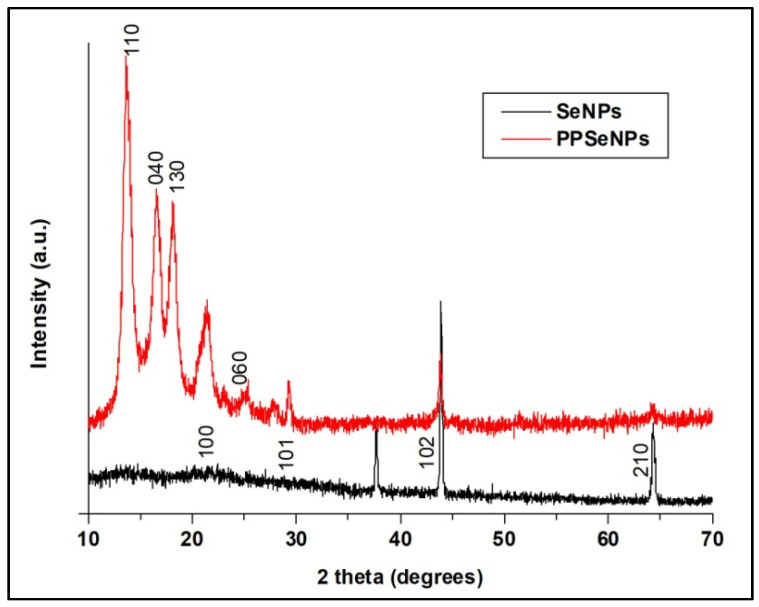
XRD patterns of the prepared SeNPs and SeNPs-PP fabric.

**Figure 3 polymers-13-02483-f003:**
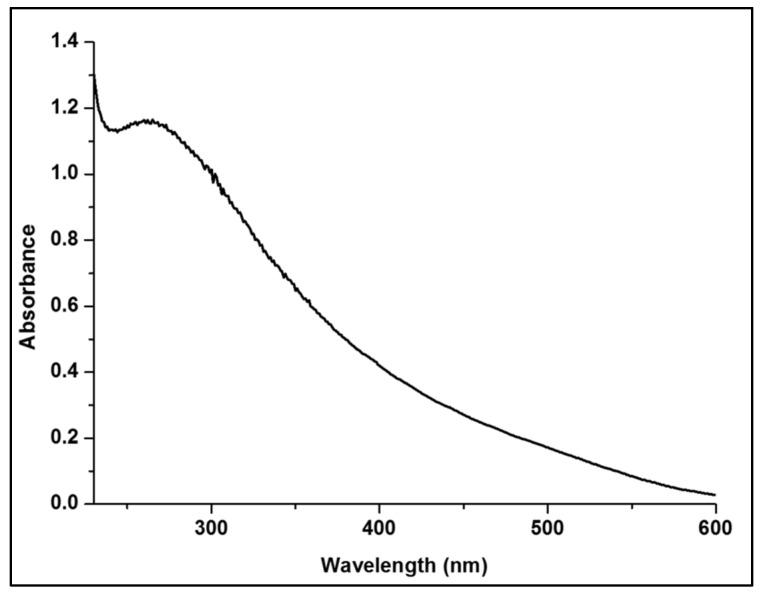
UV/Vis spectra of SeNPs at different concentrations.

**Figure 4 polymers-13-02483-f004:**
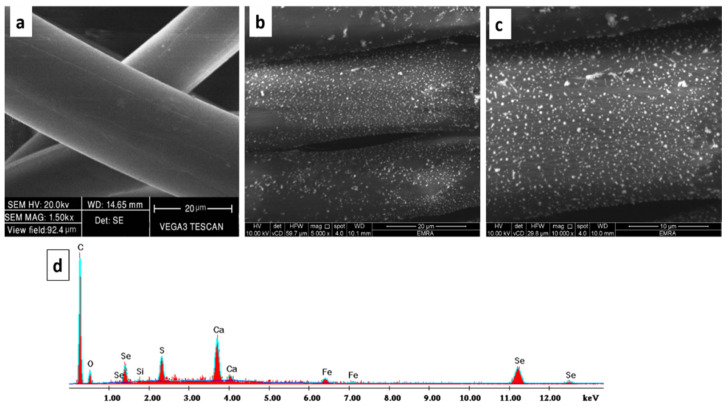
SEM micrographs of (**a**) the PP fabric, (**b**,**c**) SeNPs-PP fabric, (**d**) EDX analysis of the SeNPs-PP fabric.

**Figure 5 polymers-13-02483-f005:**
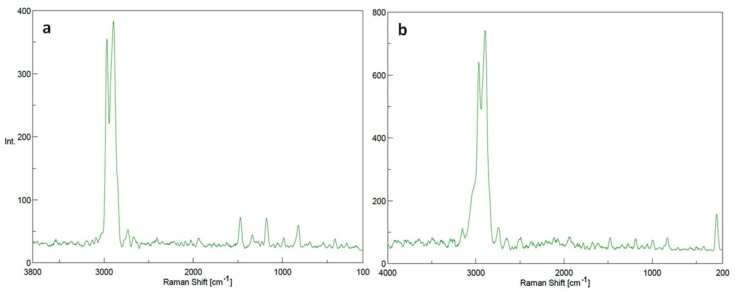
Raman spectrums of (**a**) PP fabric and (**b**) SeNPs-PP fabric.

**Figure 6 polymers-13-02483-f006:**
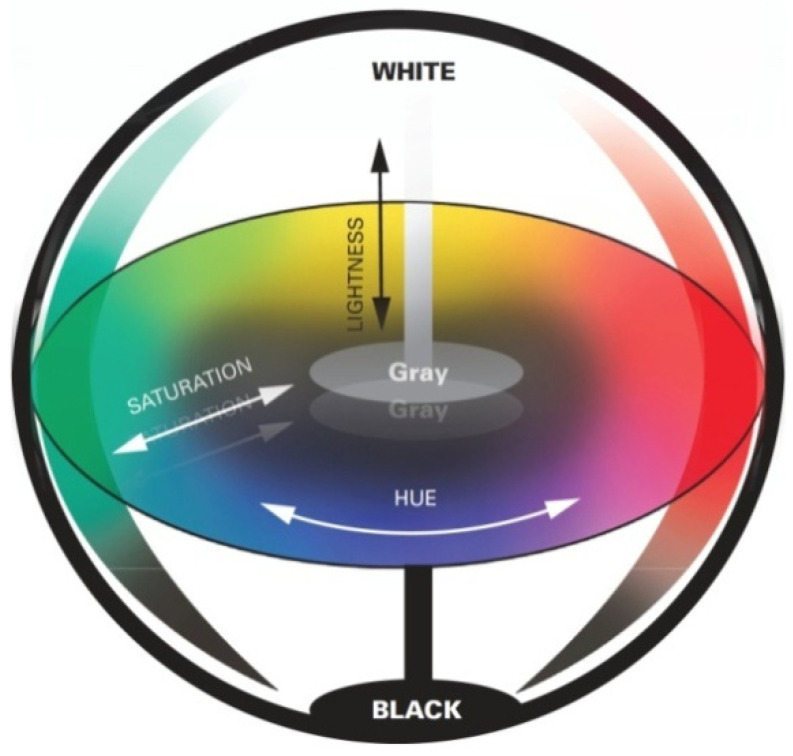
Lab colour space.

**Figure 7 polymers-13-02483-f007:**
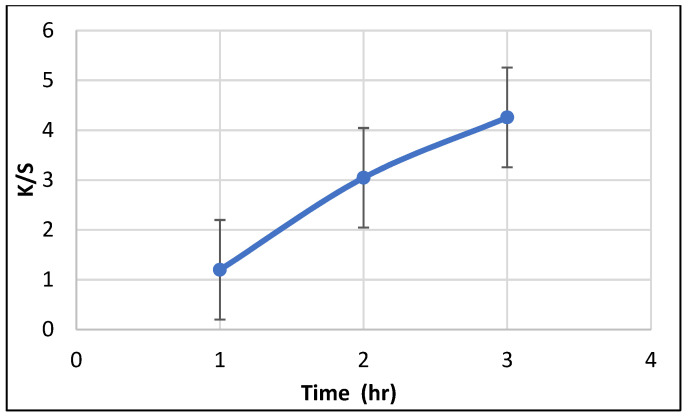
Effect of treatment time on color strength (K/S).

**Figure 8 polymers-13-02483-f008:**
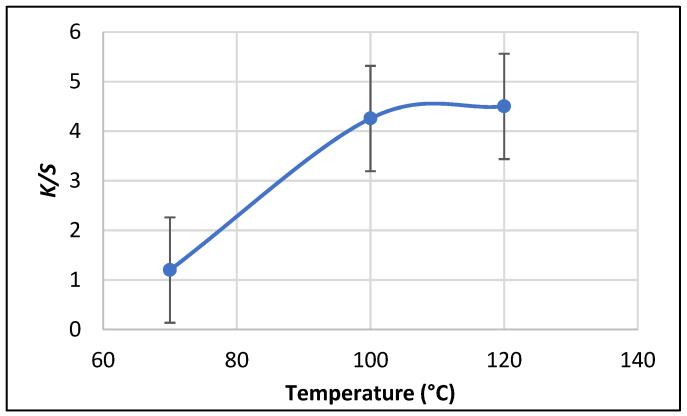
Effect of treatment temperature on color strength (K/S).

**Figure 9 polymers-13-02483-f009:**
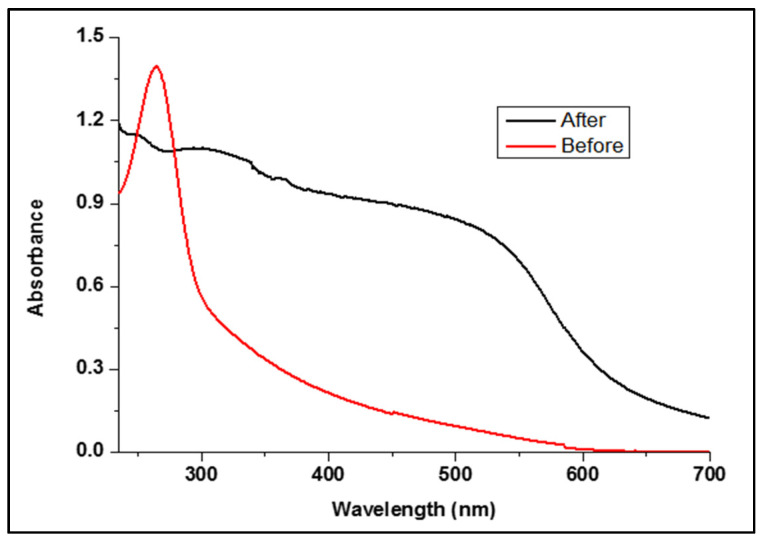
Absorbance of SeNPs before and after treatment.

**Table 1 polymers-13-02483-t001:** Optical measurements.

Type	Sample	Colour Parameters					
		L*	a*	b*	C*	h	K/S
SeNPs-PP fabric	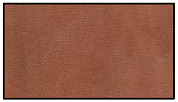	47.39	23.45	17.09	29.02	36.08	4.34

**Table 2 polymers-13-02483-t002:** Properties of the PP and SeNPs-PP fabrics under optimum conditions.

Sample	Wash Fastness		Rubbing Fastness		Light Fastness	Tensile Strength	
	St.	Alt.	Dry	Wet		Force, N	Elongation, %
PP fabric	-	-	-	-	-	1090	30.80
SeNPs-PP fabric	5	5	5	4–5	5	1102	27.36
SeNPs-PP fabric after 5 washing cycles (durability test)	5	5	5	4–5	5	-	-

**Table 3 polymers-13-02483-t003:** Clear zone (mm) of the PP and SeNPs-PP fabrics.

Substrate	Antibacterial Activity Diameter of Clear Zone (mm)			
	*Staphylococcus aureus* (G+)	*Bacillus cereus* (G+)	*Escherichia coli* (G-)	*Pseudomonas aeruginosa* (G-)
Ciprofloxacin	24	15	23	17
Tetracycline	21	14	20	15
Blank	_	_	_	_
SeNPs-PP fabric	20.9	22.7	23.2	11.3

**Table 4 polymers-13-02483-t004:** UPF of the PP and SeNPs-PP fabrics.

Sample	UVA 315–400 nm	UVB 290–315 nm	UPF Value
Blank PP fabric	35.89	29.49	3.20
SeNPs-PP fabric	0.11	0.11	920.19

## Data Availability

The data presented in this study are available on request from the corresponding author.

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
