# Peer review of "Coloration and Multi-Functionalization of Polypropylene Fabrics with Selenium Nanoparticles"

_polymers, 2021, doi:10.3390/polym13152483_

Round 1

Reviewer 1 Report

The paper presents interesting result of investigations on functionalisation of polypropylene fabric with selenium nanoparticles. By modification the change in the colour of the fabric was observed. Simultaneously, by incorporation of nanoparticles UV protection and antibacterial  properties of the fabric were improved. The paper is worthy to publish in Polymers. Before publication the major revision is required.

Generally English is poor and needs correction.

Introduction

One can discuss if selenium nanoparticles are the most promising for biomedical application.

 The section on the application of polypropylene is chaotic. The application areas and various polypropylene products  are mixed.  The application of polypropylene for hygiene products should be exposed.

Polyester and  nylon are different kind of terms. One is a chemical name, while the second is the trade name.  Equivalent term  for polyester is polyamide, not nylon.

The section on dyeing and functionalisation of polypropylene fabrics does not present the current state of art. This section should be expanded.

In the Introduction the goal of the research should be clearly presented.

Materials and methods

The characteristics of PP fabric is missing. The basic parameters of the fabric should be given.

The units [gm] and [ mM] are not known.

The producer and model of the infrared dyeing machine are missing.

The analytical methods used for characterisation of nanoparticles should be separated.

 Some details on measurements should be given (sample preparation, measurement technique,  etc.).

Results and discussion

The section is badly constructed.

The section 3.1 concerns Characterization of synthesized Se-NPs.  The paragraph 3.2.3 is not about selenium nanoparticles, but modified PP fabric. The results obtained for nanoparticles are presented in many subsections. while results obtained for fabric in many sections.

There are no scales on the surface of polypropylene fibres !!! The scales are visible for wool.

What is the origin of Ca, S and Fe on the surface of PP fibres? (Figure 3).

Where are the PP peaks on the WAXS pattern of PP fibres ? (Figure 4)

Table 1 presents only two values. What is a sense to present two values in the table?

Author Response

Comment (1)

Introduction

One can discuss if selenium nanoparticles are the most promising for biomedical application.

The section on the application of polypropylene is chaotic. The application areas and various polypropylene products are mixed.  The application of polypropylene for hygiene products should be exposed.

Polyester and nylon are different kind of terms. One is a chemical name, while the second is the trade name.  Equivalent term for polyester is polyamide, not nylon.

The section on dyeing and functionalisation of polypropylene fabrics does not present the current state of art. This section should be expanded.

In the Introduction the goal of the research should be clearly presented.

Reply

We want to thank the reviewer for these comments. The introduction has been reformulated to meet all his requests and comments. The amendments have been clarified in a separate file to track the changes in the manuscript.

Comment (2)

Materials and methods

The characteristics of PP fabric is missing. The basic parameters of the fabric should be given.

The units [gm] and [mM] are not known.

The producer and model of the infrared dyeing machine are missing.

The analytical methods used for characterisation of nanoparticles should be separated.

 Some details on measurements should be given (sample preparation, measurement technique,  etc.).

Reply

Based on your valuable advices, the characteristics of PP fabric and the model of the infrared dyeing machine were added. Also, we replaced the unit of gm with the official unit (g) and the unit of mM with mmol/l which means millimole per liter (millimolar).

Furthermore, the methods used for characterization of nanoparticles and dyed fabrics were separated as required with additional information. While, the details on measurements were added to be more informative as required also.

Comment (3)

Results and discussion

The section is badly constructed.

The section 3.1 concerns Characterization of synthesized Se-NPs.  The paragraph 3.2.3 is not about selenium nanoparticles, but modified PP fabric. The results obtained for nanoparticles are presented in many subsections. while results obtained for fabric in many sections.

There are no scales on the surface of polypropylene fibres!!! The scales are visible for wool.

What is the origin of Ca, S and Fe on the surface of PP fibres? (Figure 3).

Where are the PP peaks on the WAXS pattern of PP fibres? (Figure 4)

Table 1 presents only two values. What is a sense to present two values in the table?

Reply

the section was rearranged, where the characterization of SeNPs and the characterization of the PP fabrics were separated. In addition, based on our previous study on wool, the SEM micrograph of wool is different from that obtained for polypropylene fabric with different scales as follow:

a)     SEM micrograph of wool

b) SEM micrograph of polypropylene fabric:

-        the origin of Ca, S and Fe on the surface of PP fibres have been added.  As for XRD of PP, the peaks were illustrated as required in the XRD Figure. While the peaks at 13,66, 16.56, 18.18, and 25.38 can be observed for PP fabric, corresponding to the planes of (110), (040), (130), and (060), respectively accompanied with a reference.

Furthermore, Table 1. was deleted as required. While, the viability value was reported in the discussion.

Reviewer 2 Report

Above all, the language and style needs to be revised thoroughly. There are even words, which do not exist at all in the text - like "laterly" (page 1, 1st paragraph). Missing articles (e.g. "... were prepared using a simple chemical reduction method ..." - page 2, 3rd paragraph), missing blanks and wrong tense (e.g. "Figure 8 reports data that ..."; page 7, 2nd paragraph), superfluous commas and the like are numerous. Both the introduction and the method & material description are too short and lack reflection of common practive in PP fibre production and application.

Detail problems: 

  • The abstract is (for once!) too short and general, and the term "wi38cell line" should be replaced by a clearer expression. 
  • The sentence "In addition to the above, PP undyeable with regular dyes because they do not contain functional groups." (page 2, 3rd paragraph) is wrong or at least incomplete. While PP fibres cannot be dyed in solution processes without prior treatment (but can, if modified - see https://doi.org/10.1002/app.2175 ), melt dyeing is very well possible and performed in large quantities. UV absorbers are also available as standard additives, used in fibres as well (see https://doi.org/10.1017/S1431927618000430) - and the introduction should generally be expanded, considering for example the use of ZnO nanoparticles for a similar purpose (see https://doi.org/10.1016/j.jmst.2014.11.022).
  • The "MTT assay" cannot be assumed to be generally known among polymer scientists and must be explained briefly (page 7, 2nd paragraph). Figure 6 doesn't show anything and can be omitted. 
  • The electrical conductivity increase at a level of 1E-11 is hardly significant. 
  • Reporting detail figures in a conclusion is clearly bad style. 

Author Response

Comment (1)

Above all, the language and style needs to be revised thoroughly. There are even words, which do not exist at all in the text - like "laterly" (page 1, 1st paragraph). Missing articles (e.g. "... were prepared using a simple chemical reduction method ..." - page 2, 3rd paragraph), missing blanks and wrong tense (e.g. "Figure 8 reports data that ..."; page 7, 2nd paragraph), superfluous commas and the like are numerous. Both the introduction and the method & material description are too short and lack reflection of common proactive in PP fibre production and application.

Reply

We want to thank the reviewer for these comments. Regarding to the English language; it has been edited.  As for all comments about the introduction; it has been reformulated to meet all requests. Also the section of material and methods has been modified.

Comment (2)

The abstract is (for once!) too short and general, and the term "wi38cell line" should be replaced by a clearer expression. 

The sentence "In addition to the above, PP undyeable with regular dyes because they do not contain functional groups." (page 2, 3rd paragraph) is wrong or at least incomplete. While PP fibres cannot be dyed in solution processes without prior treatment (but can, if modified - see https://doi.org/10.1002/app.2175 ), melt dyeing is very well possible and performed in large quantities. UV absorbers are also available as standard additives, used in fibres as well (see https://doi.org/10.1017/S1431927618000430) - and the introduction should generally be expanded, considering for example the use of ZnO nanoparticles for a similar purpose (see https://doi.org/10.1016/j.jmst.2014.11.022).

The "MTT assay" cannot be assumed to be generally known among polymer scientists and must be explained briefly (page 7, 2nd paragraph). Figure 6 doesn't show anything and can be omitted. 

The electrical conductivity increase at a level of 1E-11 is hardly significant. 

Reporting detail figures in a conclusion is clearly bad style. 

Reply

We want to thank the reviewer for these comments. As for the abstract; it was edited. Regarding to cell line explanation, wi-38 cell is diploid human cell line composed of fibroblasts derived from lung tissue of a 3-month-gestation female fetus. As for the comments about the introduction, it has been reformulated again to meet all reviewer requests. Also, the MTT abbreviation was explained in the materials and methods followed by MTT abbreviation in a bracket to be known for the reader.  Additionally, Figure 6 was deleted as required from the manuscript, while the viability percentage was added in the text.

In concerning with electrical conductivity value, we agreed with reviewer about; the electrical conductivity increase at a level of 1E-11 is hardly significant; so we changed the description through text to that the increase was a slight increase.  As for reporting details of figures in a conclusion, the details were deleted.
